# Melanoma Clinical Decision Support System: An Artificial Intelligence-Based Tool to Diagnose and Predict Disease Outcome in Early-Stage Melanoma Patients

**DOI:** 10.3390/cancers15072174

**Published:** 2023-04-06

**Authors:** Jose Luis Diaz-Ramón, Jesus Gardeazabal, Rosa Maria Izu, Estibaliz Garrote, Javier Rasero, Aintzane Apraiz, Cristina Penas, Sandra Seijo, Cristina Lopez-Saratxaga, Pedro Maria De la Peña, Ana Sanchez-Diaz, Goikoane Cancho-Galan, Veronica Velasco, Arrate Sevilla, David Fernandez, Iciar Cuenca, Jesus María Cortes, Santos Alonso, Aintzane Asumendi, María Dolores Boyano

**Affiliations:** 1Dermatology Service, Cruces University Hospital, 48903 Barakaldo, Spain; 2Biocruces Bizkaia Health Research Institute, 48903 Barakaldo, Spain; 3Dermatology Service, Basurto University Hospital, 48013 Bilbao, Spain; 4TECNALIA, Basque Research and Technology Alliance (BRTA), 20850 Gipuzkoa, Spain; 5Department of Cell Biology and Histology, University of the Basque Country/EHU, 48940 Leioa, Spain; 6Department of Psychology, Carnegie Mellon University, Pittsburgh, PA 15213, USA; 7Ibermática Innovation Institute, 48170 Zamudio, Spain; 8Pathology Service, Basurto University Hospital, 48013 Bilbao, Spain; 9Pathology Service, Cruces University Hospital, 48903 Barakaldo, Spain; 10Department of Genetics, Physical Anthropology and Animal Physiology, University of the Basque Country/EHU, 48940 Leioa, Spain; 11NorayBio, 48160 Zamudio, Spain; 12IKERBASQUE, The Basque Foundation for Science, 48009 Bilbao, Spain

**Keywords:** melanoma, biomarkers, diagnosis, prognosis, machine learning, deep learning, artificial intelligence, metastasis, disease-free, risk factors

## Abstract

**Simple Summary:**

Early diagnosis and accurate prognosis is essential to personalize treatment and improve the survival of melanoma patients. We report here a new tool that can improve the early diagnosis of melanoma through the use of epiluminescence dermatoscopy and deep learning image analysis. By employing artificial intelligence algorithms to analyze simple serological and histopathological biomarkers, the risk of metastasis and the disease-free interval of melanoma patients can be accurately predicted. This low-cost Melanoma Clinical Decision Support System represents an effective tool to help clinicians manage melanoma patients.

**Abstract:**

This study set out to assess the performance of an artificial intelligence (AI) algorithm based on clinical data and dermatoscopic imaging for the early diagnosis of melanoma, and its capacity to define the metastatic progression of melanoma through serological and histopathological biomarkers, enabling dermatologists to make more informed decisions about patient management. Integrated analysis of demographic data, images of the skin lesions, and serum and histopathological markers were analyzed in a group of 196 patients with melanoma. The interleukins (ILs) IL-4, IL-6, IL-10, and IL-17A as well as IFNγ (interferon), GM-CSF (granulocyte and macrophage colony-stimulating factor), TGFβ (transforming growth factor), and the protein DCD (dermcidin) were quantified in the serum of melanoma patients at the time of diagnosis, and the expression of the RKIP, PIRIN, BCL2, BCL3, MITF, and ANXA5 proteins was detected by immunohistochemistry (IHC) in melanoma biopsies. An AI algorithm was used to improve the early diagnosis of melanoma and to predict the risk of metastasis and of disease-free survival. Two models were obtained to predict metastasis (including “all patients” or only patients “at early stages of melanoma”), and a series of attributes were seen to predict the progression of metastasis: Breslow thickness, infiltrating BCL-2 expressing lymphocytes, and IL-4 and IL-6 serum levels. Importantly, a decrease in serum GM-CSF seems to be a marker of poor prognosis in patients with early-stage melanomas.

## 1. Introduction

Cutaneous melanoma is the deadliest type of skin cancer, although survival is currently being enhanced due to prevention strategies and diagnosis at early stages of tumor development. Nevertheless, the treatment of advanced melanoma remains limited [1], and early detection and surgical treatment still constitute the best means to improve the outcome of melanoma patients. The diagnosis of a primary melanoma is achieved by visual examination of suspicious lesions followed by biopsy. Moreover, the acquisition of clinical images (photographs of the lesions) can now be complemented by dermatoscopy examination, currently an important tool to help dermatologists distinguish melanomas from other pigmented skin lesions [2]. Consequently, software based on artificial intelligence (AI) is being developed to automate the analysis of images in order to facilitate early diagnosis of melanoma, and to help dermatologists in both hospital environments and in primary care consultations [3].

The prognosis of melanoma is defined mainly through histopathological features, where Breslow thickness and the presence of ulceration are considered the most important hallmarks [4,5]. However, both these features are most relevant when the neoplastic lesion is already at an advanced stage. Around 20% of patients diagnosed with early-stage melanoma (stages I and II, according to the AJCC) develop metastasis in the following 5 years [6], and, significantly, some stage II patients have worse survival than stage III patients. This phenomenon can be explained by the heterogeneity in the changes during the evolution of this neoplastic disease, both the genetic alterations and those related to melanoma cell plasticity [7]. In this context, there is intense activity aimed at discovering new biomarkers that can offer accurate predictions. These biomarkers must have the capacity to stratify melanoma patients—for instance, based on proteomic, lipidomic, NGS, and/or microbiome profiles, and ideally through noninvasive techniques [8]. However, there are currently no melanoma risk stratification tools that have been well validated or that are widely used.

In this context, for many years we have focused on classifying early-stage melanoma patients on the basis of serological and novel histopathological biomarkers that can be assessed at the time of melanoma diagnosis and initial surgery, thereby identifying profiles that are most likely to develop to more advanced stages and enabling more effective treatments for metastatic disease to be implemented [9,10,11]. Thus, in independent studies we previously studied, on the one hand, cytokines in the serum of patients [9] and, on the other hand, histopathological biomarkers in melanoma biopsies [9,10]. We found that cytokines such as IL-4, IL-6, and GM-CSF are significantly related to a bad prognosis of patients [9]. In a retrospective study, we have also demonstrated the relationship between RKIP and Pirin protein expression in melanoma biopsies and malignant progression [10,11]. In the present work, we wanted to know the predictive value of these markers taken all together in an integrative study using AI. For that, here, serum levels of interleukins (ILs) such as IL-4, IL-6, IL-10, and IL-17A, IFNγ (interferon-γ), GM-CSF (granulocyte and macrophage colony-stimulating factor), TGFβ (transforming growth factor β), and DCD (dermicidin) were assayed, and the RKIP, PIRIN, BCL2, BCL3, MITF, and ANXA5 proteins were assessed by immunohistochemistry (IHC) in melanoma biopsies from a group of melanoma patients. Moreover, the presence of infiltrating lymphocytes and macrophages in the tumor sections was also quantified. By integrating these results and taking into consideration demographic and clinical data, we designed models based on machine learning (ML) techniques that are capable of accurately classifying early-stage melanoma patients with a high or low risk of developing metastasis and predicting the metastasis disease-free period in months. The Melanoma Clinical Decision Support System (CDSS) described here includes a diagnostic module based on deep learning (DL) analysis of clinical and dermatoscopy images and a prognostic module based on the ML analysis of the other markers, integrating the expression of all the serological and immunohistopathological biomarkers analyzed along with the relevant clinical information. This information could be used clinically to determine the probability of developing metastasis during follow-up and to make a decision about whether early-stage melanoma patients should receive adjuvant therapy to prevent metastasis. Thus, the Melanoma CDSS represents a new and useful diagnostic tool that can also be used to predict the probability of recurrence and survival, as well as to identify appropriate candidates for adjuvant treatment.

## 2. Materials and Methods

### 2.1. Patients

This is a prospective and longitudinal study based on clinical, serological, and molecular biomarker detection in 196 melanoma patients diagnosed at the Basurto and Cruces University Hospitals (Euskadi, Spain) between 1990 and 2016. The last revision of the disease follow-up or the status of the patients was updated in December 2019 (Table 1). The criteria for patient inclusion were a histologically confirmed diagnosis of malignant melanoma; no treatment except primary surgery (including wide local excision); and no infection based on the clinical evaluation and absence of increased parameters of infection in the blood. Biopsies of suspicious lesions were analyzed by a pathologist, and those patients with a positive diagnosis of melanoma underwent a second surgery to achieve a wide local excision. Patients diagnosed with stage III or IV melanoma were referred to the hospital’s Oncology Unit, whereas stage I or II patients (henceforth “early-stage melanomas”) remained under the supervision of the Dermatology Unit. Upon removal of the primary tumor, clinical check-ups of the patients with early-stage melanomas were scheduled every 3 months for the first 2 years of the follow-up and every 6 months thereafter, until a 5-year follow-up had been completed. Annual revisions were then scheduled up to the 10th year post-surgery. Any patient who developed a metastasis during the follow-up period was examined again every 3 months for 2 years after the metastasis had been diagnosed and treated. The presence or absence of metastasis was assessed in all patients by physical examination, as well as through laboratory and radiological testing (X-rays and/or computed tomography (CT) scanning). Some patients underwent sentinel lymph node biopsy, although this was not a generalized procedure. Disease stages were classified according to the AJCC eighth edition [4], and the clinical and diagnostic data for each patient were collected retrospectively from centralized electronic and/or hard copy medical records. For some of the statistical prediction analysis, only early-stage melanoma patients (I and II stages) were included. Patients who did not develop metastasis in the first 5 years of the follow-up were grouped as “disease-free.”

The study was conducted in accordance with the Declaration of Helsinki regarding research on humans, and it was approved by the Euskadi Ethics Committee (No. PI 16–99), with written informed consent obtained from all the subjects.

### 2.2. Serological and Molecular Tumor Biomarkers

The serum samples collected from the patients were stored at −80 °C, and Formaldehyde Fixed Paraffin Embedded (FFPE) melanoma biopsies were held at the Basque Biobank until use (https://www.biobancovasco.org/, accessed on 3 April 2023). Serological determination of the cytokines (IL-4, IL-6, IL-10, IL-17A, IFNγ, GM-CSF, and TGFβ) and DCD was performed on serum from 196 melanoma patients. The IHC biomarkers (RKIP, PIRIN, BCL3, BCL2, MITF, and ANXA5) were studied in the same patients as the serum biomarkers, and the presence of infiltrating lymphocytes and macrophages in the histological melanoma sections was also quantified. The characteristics of the patients are shown in Table 1, and a summary of the previous studies performed to discover new melanoma risk factors to include in the Melanoma-CDSS is shown in Appendix A [9,10,11]. Serological markers were quantified using the MILLIPLEX MAP Kit, Human High Sensitivity T Cell Magnetic Bead Panel (EMD Millipore Corporation, Darmstad, Germany). Serum DCD was quantified with an ELISA Kit (Cusobio Biotech Co. Ltd., Houston, TX, USA) according to the manufacturer instructions. For the IHC analysis, sections from FFPE blocks (4 μm thick) were first subjected to antigen retrieval in citrate buffer [pH 6.1] and steam-treated for 105 min, and immunohistochemistry was then performed by probing them with antibodies against RKIP, PIRIN, BCL3, BCL2, MITF, and ANXA5 (Thermo Fisher Scientific, Waltham, MA, USA). The staining of these proteins are shown in the Figure 1 in representative primary melanoma biopsies. Antibody binding was assessed using the EnvisionTMG|2 System/AP Kit (Dako Corporation, Glostrup, Denmark), counterstaining the slides with hematoxylin and obtaining images on a NanoZoomer S210 Digital slide scanner (C13239-01, Hamamatsu Photonics, Japan). The staining intensity was evaluated as negative, low, or high based on independent examination by two observers. Discordant assessments were reviewed jointly to obtain a conclusive consensus evaluation. The presence of infiltrating lymphocytes expressing BCL2 (L-BCL2^+^) was also analyzed and quantified as negative, low, or high L-BCL2^+^ expression.

### 2.3. The Melanoma Clinical Decision Support System (Melanoma CDSS)

The Melanoma CDSS is an application capable of extracting useful information to help clinicians reach a diagnosis and define the prognosis of melanoma patients based on AI models. The Melanoma CDSS integrates two modules that send the necessary information to each AI model, and it collects the output information before visualizing, storing, and displaying it. The Melanoma CDSS includes all the user interface (UI) functionalities and different screens, from controlled access to patient selection for the diagnosis to the prognosis modules, and it connects and integrates the different processing modules. The Melanoma CDSS also takes into account security and confidentiality issues, through data encryption, password management, user logging, etc. Moreover, it includes data I/0 management for both massive batch upload and one-by-one data upload, also storing the data for later use by clinicians. The functional version of the CDSS UI integrates clinical patient information, biopsy-derived information (serum biomarkers, IHC biomarkers, etc.), a diagnostic module based on clinical and dermatoscopy images, and a prognostic module for metastasis prediction based on molecular data and disease-free time prognosis (in months) that employs multimodal data. Both the diagnostic and prognostic modules are assisted by AI models.

In a scheme of the system’s workflow, how the diagnostic and prognostic modules make use of the available information (images and data) is illustrated (Figure 2). For each new patient, suspicious lesions are studied with a dermatoscope, and the images collected are analyzed automatically by the “diagnosis module” of the CDSS, offering a suggested diagnosis. For lesions suspected to be malignant, a biopsy is taken, and the serum and IHC biomarkers are examined to achieve a pathological diagnosis. The “prognosis module” of the CDSS then uses this information, along with additional data (sex, age at diagnosis, lesion location, disease-free survival, etc.; Figure 2), to provide a probability of metastasis and a prediction of patient survival.

#### 2.3.1. Diagnostic Module

The computer-aided diagnosis (CADx) tool has been developed to use clinical and dermatoscopy images. Two separate AI models were developed, considering that both types of images might not be available for all the lesions, and complementary diagnostic predictions were calculated to reinforce and improve current clinical diagnosis. In the “diagnosis module”, AI models are based on DL approaches, the current gold standard for medical image processing. DL-based models require large amounts of data for training, validation, and testing, which is why external datasets are commonly used for the initial configuration of these models (i.e., to set the initial weights of the network). Data augmentation and imbalance techniques are also usually incorporated into the models in order to deal with the heterogeneity of the data and to maximize the benefits of the training process.

The AI models developed make use of DL convolutional neural networks (CNNs) [12,13] for the automatic characterization and classification of images, where most recent examples’ performance [14] demonstrate the technical feasibility as a tool in the daily clinical routine, although still challenging due to different limitations [15,16,17]. In the case of clinical images, a custom model was pre-trained on a database of over 1600 clinical images of nevus, melanoma, and seborrheic keratosis obtained from the public DermNet initiative [18], and these were used for training, validation, and testing of the proposed algorithm [19]. The target problem is the classification of lesions into two classes: melanoma or nonmelanoma (nevus and seborrheic keratosis). Retraining (fine-tuning) of the model was then performed with the images taken from the 196 cases collected, splitting the sample into training, validation, and test sets to evaluate the model. In addition, in the second model, a similar approach was adopted for dermatoscopy images using the dataset proposed in the ISIC 2017 challenge [20] (see Figure 3). These images present differences inherent to the device used and related to lighting, colors, shadows, etc., all of which can affect the performance of the algorithm. For this reason, the second model proposed includes a correction method for illumination and a lesion segmentation procedure, which proved to be very useful [21]. In this case, the target classification problem is twofold: melanoma versus others, or seborrheic keratosis versus others.

The performance metrics used from DL analysis for the diagnosis of primary cutaneous melanoma were the sensitivity and the specificity. Sensitivity, also known as the hit rate or the true positive rate (TPR), refers to the ability of the model to correctly classify a lesion in accordance with the lesion’s diagnosis, i.e., the probability that a true melanoma lesion is classified as such. The specificity, also known as the selectivity or true negative rate (TNR), refers to the ability of the model to correctly classify healthy cases as not lesions, i.e., the probability that a lesion that is not a melanoma is classified as a nonmelanoma lesion.

#### 2.3.2. Prognostic Module

In this section, a prototype advanced ML algorithm was designed to establish skin cancer prognosis, having first defined two different objectives to predict disease-free survival (more or less than 5 years) and metastasis (yes or no). A dataset was available with information from 196 patients and covering 28 attributes regarding clinical and experimental data (see the attributes listed in Figure 2). The flowchart of the dataset is shown in Appendix A. Different decision tree (DT) [22] algorithms and support vector machines (SVMs) with different kernel functions (dot, radial, polynomial, neural, and ANOVA) [23] were developed to define the best model to predict both metastasis and disease-free survival (see Appendix A for a list and brief description of the algorithms used in this analysis). The different processes and their associated subprocesses, as well as the models, were developed using RapidMiner software [24]. To train the models, a 10-fold cross-validation using stratified sampling was obtained to test the ability of the ML algorithms to predict new data efficiently [25]. The best model was obtained using the RapidMiner DT model, and the parameters selected in each algorithm used in the prognosis module are shown in Appendix A.

The metrics used from the machine learning analysis to predict metastasis and disease-free survival were specificity (the proportion of patients predicted as no metastasis who are actually no metastasis) and sensitivity (the proportion of truly metastatic patients in the model relative to the actual metastatic patients in the data). In this case, we have also included precision, as the ratio of the true metastatic patients in the model to the total of the metastatic patients predicted by the model, and the F1 score, which accounts for both precision and sensitivity; it is the harmonic mean (average) of the precision and sensitivity.

## 3. Results

### 3.1. AI for the Diagnosis of Primary Cutaneous Melanoma

As indicated above, two different AI models have been developed, and the lesion classification model metrics based on clinical images are shown in Figure 4. The relationship between sensitivity and specificity metrics enables the best cut-off threshold to be identified (Figure 4A), and, as such, the three dashed vertical lines in blue defined in the graph are the cut-off thresholds chosen from the comparative analysis (Figure 4A), whereby a 96.82% sensitivity and 75.41% specificity was obtained as the best “sensitivity cut-off”, or 91.72% sensitivity and 91.72% specificity for the “sensitivity = specificity cut-off”, or 86.62% sensitivity and 93.44% specificity as the best “specificity cut-off” depending on the cut-off value (Figure 4B). This performance surpassed previously reported metrics of clinicians’ assessments, demonstrating the relevance and importance of the information provided by classical clinical images when analyzed using DL algorithms.

On the other hand, when using dermatoscopy images, an average 84% sensitivity and 69% specificity was achieved by the model for the defined tasks. The model and implemented approach are defined in detail in [21]. For the dermatoscopy model, the dataset employed was more challenging as it included images with different types of illumination, stickers, or pen marks on the lesions, rulers, etc.—elements that, to some extent, distract the model from the targeted classification problem. Strategies to overcome these problems, especially color correction methods, have been implemented in the proposed approach. The more conservative results obtained with the model trained over dermatoscopy images demonstrate the relevance of the macroscopic clinical features of the lesions identified by the model trained over clinical images, suggesting that lesions should be evaluated as a whole for better diagnosis. Both types of images seem to provide complementary information, and modern DL approaches used to build hybrid models could be adopted to improve the performance, adding more clinical information that describes the main features of the lesions.

### 3.2. AI to Predict Metastasis and Disease-Free Survival

The performance of different models developed to predict if a patient will suffer metastases during the follow-up or not was also assessed (Figure 5). Most of the models displayed similar accuracy in terms of the proportion of correctly classified patients among all the patients, and some of them had very good specificity, although those models with very high specificity had very low sensitivity (Appendix A). These models were random forest DT and SVM algorithms with radial basis function as a kernel with 0% sensitivity, and SVMs with a neural network as a kernel that displayed only 2% sensitivity. This means that these models were not able to predict when patients develop a metastasis, probably due to the unbalanced datasets used in which there were many more patients without metastasis (n = 131) relative to the metastatic patients (n = 65).

According to the F1 score, the best model is that obtained using the RapidMiner implementation of the DT algorithm, with an 81% F1 score. This model achieved 74% sensitivity, 89% precision, and 88.24% accuracy (see the DT model in Figure 5A and Appendix A).

Although RapidMiner DT is the best model to predict metastasis according to its F1 score, other models have also a good F1 score as SVM with ANOVA as kernel (see Appendix A) or decision stump decision tree to predict metastasis in patients without metastasis at the time of data collection (see Appendix A). On the other hand, other models have very poor F1 score and sensitivity as SVM with neural as kernel or random forest to predict metastasis in the 196 patients (see Appendix A) or SVM with neural or radial kernel to predict metastasis in patients without metastasis at the time of data collection (see Appendix A).

A DT model is a tree-like model that serves as a decision support tool, visually displaying decisions and their potential outcomes. The paths from the root to the leaves represent classification rules; the red bar at the leaf is the proportion of patients with metastasis, while the blue bar is the proportion of patients without metastasis. In the model obtained, the first attribute evaluated is disease-free survival. If the value of this attribute is less than 21.5 months, then the patient will be predicted to have had a metastasis with a probability of 94.12% ((48/(48 + 3)) × 100). According to the model, the most relevant attributes to obtain the rules are disease-free survival, Breslow index, L-BCL2^+^, and serum IL-4 levels (pg·mL^−1^).

Furthermore, in order to obtain a model to use in a predictive way with newly diagnosed patients rather than to make a descriptive study of the available information afterwards, a new model is trained with the original 196 patients using the RapidMiner decision tree algorithm (since this has been the best algorithm in the two previous approaches), but without the disease-free survival attribute. For the model trained using the 196 patients, the tree obtained is shown in Figure 5B (rules in Appendix A). As can be seen, the attributes used to predict metastasis in a patient are the Breslow index, AJCC, and IL-6 serum levels. Compared with the model using DFS, in this case, the performance metrics decrease, although the accuracy of the model is 75%.

New models were obtained by selecting those patients without metastasis at the time of data collection from the dataset of 196 patients (patients diagnosed as stage I and II according to AJCC eighth edition). In order to conduct a useful and realistic study, those with less than a two-year follow-up and no metastasis were removed from the original dataset (196 patients) to avoid including patients for whom it was not yet clear whether they will develop metastases because their disease-free time is so short. A total of 47 patients were removed, obtaining a new dataset with 149 patients. The goal was to obtain a model to predict if a patient will develop metastasis or not before the cancer reached an advanced stage. When the same metrics were assessed as for the previous models, the same algorithms (random forest DT, SVM with radial basis function as a kernel, and SVM with neural function as a kernel) had 0% sensitivity (Figure 6A and Appendix A). Again, this implies that these models predict all patients as no metastasis; therefore, the models are not able to predict the development of metastasis. This is again probably due to a lower proportion of patients without metastasis in the dataset (43 patients with metastasis and 106 patients with no metastasis), as in the previous model with 196 patients.

According to the F1 score, the best model was the RapidMiner DT, with a F1 score of 73%, 87% accuracy, 63% sensitivity, and 96% specificity (Figure 6A). According to the model, the most relevant attributes were the same as those in the previous model in conjunction with GM-CSF serum levels, which did not appear in the previous model. Moreover, the threshold values to generate the rules were also the same as in the previous model (Appendix A). From the model, it can be concluded also that for those patients with GM-CSF serum levels ≤250.7 pg·mL^−1^ and >10.7 pg·mL^−1^, metastasis development would be predicted within the first 21.5 months after excision of the primary melanoma.

As before, in the case of patients diagnosed as stage I and II of AJCC, without metastasis at the time of data collection, the model obtained is shown in Figure 6B, where AJCC, Breslow index, and IL-4 and IL-6 serum levels are attributes used to predict metastasis. As in the previous case, the performance metrics compared with the model with DFS decrease. However, the Breslow index and IL-4 are attributes used by both models, which implies that they are relevant variables in the metastasis.

### 3.3. Application

The Melanoma CDSS for skin cancer manages the security issues related to the application (e.g., users, passwords, logs) and handles all the clinical information regarding the hospital, doctor, lesions, patient, their visits, diagnosis, prognosis, etc., as well as the relationships among these features, unifying all this information. The incorporation of patient information can be achieved using an encrypted Excel file or directly through the application itself. Thus, the Melanoma CDSS for skin cancer is an application with a two-fold usage:

Diagnosis of skin lesion in the first clinical session with the patient. The Melanoma CDSS will guide the dermatologist through the diagnostic module, which will show the dermatologist all the clinical information stored in the clinical database, as well as all the information from the image-based diagnostic module. The image diagnosis module will show the tool’s predicted diagnosis and its accuracy.

Prognosis of a biopsied skin lesion after surgery. Once the patient has been diagnosed with skin cancer and after surgery, the medical protocol determines a series of tests that should be performed on the patient. Once this information is stored in the Melanoma CDSS, the dermatologist can check the values and may also use the molecular-database-driven prognostic module to obtain a prediction of the patient’s life expectancy. Like the image diagnosis module, this tool will be used at the request of the clinician. The clinician may also use the multiple-sources prognostic module to obtain additional information about the patient’s predicted classification into one of various categories related to survival, gaining information as to why the patient has been classified in this way, and regarding other patterns and relationships through the relevant variables available in the dataset, and from other patients (see Appendix A for screenshots from the Melanoma CDSS application modules).

## 4. Discussion

Despite clinical staging guidelines, the heterogeneous nature of melanoma makes its diagnosis and prediction challenging, even for experienced dermatologists. Indeed, the wide range of morphologies among skin lesions makes it difficult to distinguish melanomas from other pigmented skin lesions. The Melanoma CDSS presented here is a bioinformatics application based on AI algorithms capable of extracting information that may be useful to improve the diagnosis and prognosis of patients with cutaneous melanoma. It makes use of clinical, demographic, and molecular data obtained by quantifying serum cytokines and other histopathological biomarkers in melanoma biopsies from patients. The Melanoma CDSS provides quick and insightful answers to clinician and user questions, presented in an easy-to-understand visual format.

The initial evaluation of dermatological lesions, specifically melanomas, relies on the identification of ABCDE features [26], whereby lesion asymmetry, borders, color, diameter, and evolution are studied. Lesion features can be studied by eye or assisted by dermatoscopy, but the final diagnosis relies on biopsy and histopathological diagnosis, which takes 3 or 4 weeks to complete. For this reason, we have developed a computer-aided diagnostic CADx tool to assist clinicians in the diagnostic procedure when examining a lesion. The development of CADx applications in dermatology is an area of research that has achieved significant relevance in recent years due to the increasing number of appointments in primary care and at dermatology units. The latest advances in AI, and more specifically in DL, represent a landmark in modern medicine. Solutions based on these tools have been seen to perform at an expert level [12,27], paving the way to adopt such CADx solutions in clinical practice, and although there are still several challenges that remain to be overcome [28], great progress continues to be made. As demonstrated here, promising results can be obtained by applying DL tools to clinical and/or dermatoscopy images [28,29]. As a result, we believe that feeding a combination of both these sources of information into AI models can significantly improve their performance as they provide complementary information. DL strategies are continuously evolving, and they offer improved performance and a new range of possibilities in biomedicine—for example, facilitating continued learning or the possibility to combine not only images but also clinical and genomic or molecular data in hybrid/multi-source models.

Cutaneous melanoma is a genetically heterogeneous disease in which different patient subgroups are associated with different outcomes. Bioinformatics tools have been widely used to analyze Next Generation Sequencing (NGS) data in order to identify gene mutations potentially associated with the etiopathogenesis of melanoma [30,31,32]. Moreover, the study of the integrated properties of melanoma, its tissue microenvironment, and immune invasion is guided by classical histopathology, while the profiling of protein or RNA expression in particular has revealed some molecular features associated with tumor development that involve paracrine cytokine signaling [33,34]. Thus, the spatial landscape of primary melanoma progression is being enhanced by the use of multiplex analyses [35,36]. Clinical and histopathological risk factors are widely used to define prognosis, for risk stratification, and to help achieve personalized treatment for cutaneous melanoma. Systemic adjuvant therapies for stage III and IV melanoma are now widely used following the surgical resection of the advanced melanoma. However, adjuvant treatment is not usually recommended for stage I and II melanoma, even though the mortality rate at 10 years may be relatively high [37]. Indeed, in an analysis of the gene expression profiles of 523 primary melanomas, 70% of stage I and II patients developed distant metastasis [34].

There are currently no well-validated or widely used melanoma risk stratification tools. Here we present an inexpensive method based on histopathological and serological biomarkers, information that, once fed into AI models, can be used to achieve an accurate diagnosis and a precise prognosis at early stages of melanoma. Moreover, our predictive AI analytics may help oncologists make treatment decisions. Along with the Breslow thickness, the Melanoma CDSS includes features such as the relative number of tumors infiltrating L-BCL2^+^ and serum levels of IL-4 or IL-6, all of which have been reported to be important risk factors for melanoma progression [9]. The Breslow thickness alone has been considered a malignant risk factor [34], but an important increase in the predictive power of the Breslow Index was achieved by combining it with data regarding infiltrating L-BCL2^+^ and serum levels of IL-4 and GM-CSF. Models obtained using RapidMiner DTs show that the Breslow index, infiltrating L-BCL2^+^, and IL-4 levels are associated with a poor prognosis, whereas a decrease in GM-CSF serum levels directly identifies early-stage melanoma patients in whom the disease-free survival is less than 21.5 months. However, the mentioned models use DFS as a main predictive attribute, so their clinical use is limited. Metastasis prediction models where DFS has not been considered have potential for real clinical use and show that Breslow index, stage, and serum levels of IL-4 and Il-6 are markers of poor prognosis. Although the performance metrics decrease compared with the initial models including DFS, Breslow index, and IL-4 keep as attributes used in both models, which implies that they are relevant variables in metastasis development. The melanoma microenvironment contains stromal cells and immune cells, such as T- or B-lymphocytes, NK cells, or tumor-associated macrophages that can secrete cytokines and that might play a role in inhibiting or promoting tumor progression [32]. The activation of infiltrating lymphocytes could be analyzed by determining the expression of certain antigens using specific monoclonal antibodies. Surprisingly, BCL-2 expression was detected here in both melanoma cells and in the tumor-infiltrating lymphocytes. Moreover, the algorithm revealed that the expression of BCL-2 in the melanoma-infiltrating lymphocytes constitutes an important risk marker to predict metastasis. BCL-2 is an anti-apoptotic member of the B-cell lymphoma-2 family of proteins [38]. Lymphocytes with high BCL-2 expression could be reflecting the activation or a proliferative state of these cells, and in lymphoma BCL-2 expression is considered a potential marker of poor prognosis [39]. In the melanoma microenvironment, it remains unclear if the presence of infiltrating lymphocytes is associated with a good or bad prognosis [40]. Infiltrating lymphocytes are a functionally heterogeneous group of cells that could differentiate to induce an anti-tumor response or to inhibit the immune response against melanoma cells [41]. In the group of melanoma patients studied, although the phenotype of the infiltrating lymphocytes was not determined, a high number of infiltrating L-BCL2^+^ and levels of IL-4 > 48.5 pg·mL^−1^ are associated with metastatic progression (86.82% accuracy), suggesting that an immunosuppressive response is being induced in melanoma cells.

Finally, GM-CSF is a hematopoietic growth factor that fulfills a fundamental role in macrophages and granulocyte differentiation and that has been described as a promotor [42] or inhibitor [43] of tumor progression. Although we did not find significant differences in the number of infiltrating macrophages in melanoma biopsies here, the results are consistent with studies describing an anti-tumor activity of GM-CSF [44], albeit exclusively in the early stages of melanoma. Thus, a decrease in GM-CSF serum levels in patients with stage I and II melanomas appears to be associated with the development of metastasis.

To summarize, sensitive and accurate models can be obtained using different algorithms and two different datasets: one with a total 196 patients, and the other with 149 patients after removing the patients with less than a two-year follow-up who were nonmetastatic at the time of diagnosis. In both cases, the best results are obtained using a RapidMiner DT algorithm, and similar trees with some equivalent rules were used in both models to achieve prognosis and to predict metastasis. The most accurate and sensitive model is obtained with the dataset containing all 196 melanoma patients, and both models predict metastasis (that with all patients and with patients with early-stage melanoma) based on the attributes of Breslow thickness, infiltrating L-BCL2^+^, and serum IL-4 and IL-6 levels. Importantly, a decrease in serum GM-CSF levels seems to be a marker of poor prognosis only in patients with early-stage melanoma.

## 5. Conclusions

Cutaneous melanoma is a heterogeneous neoplasia with many patient subgroups essentially established on the basis of the appearance of metastases during follow-up. AI solutions for melanoma diagnosis have been designed here and, importantly, to also predict the appearance of metastasis in early-stage melanoma patients. Along with Breslow thickness, infiltrating L-BCL2^+^, and serum IL-4, IL-6 and GM-CSF levels appear to be relevant risk factors for melanoma progression.

## 6. Patents

European patent No. EP3051291 (EP14796149.4). “Method for diagnosis and prognosis of cutaneous melanoma”. University of the Basque Country (UPV/EHU).

## Figures and Tables

**Figure 1 cancers-15-02174-f001:**
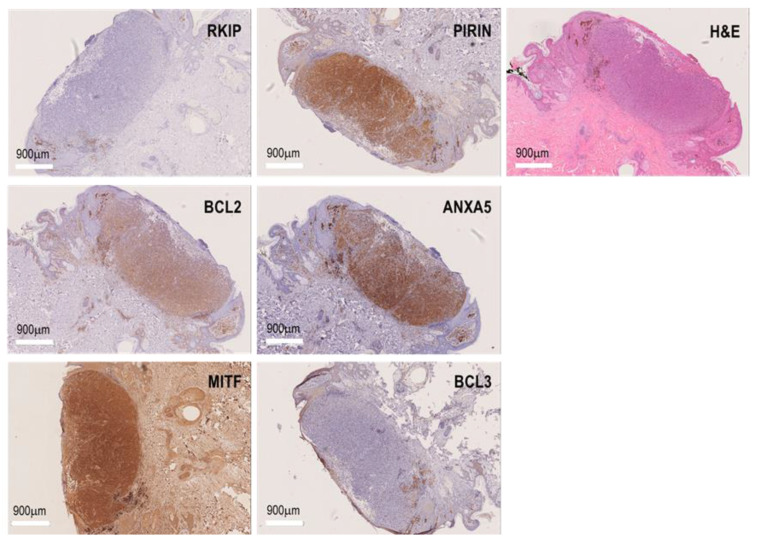
Biomarker expression in FFPE biopsies of melanomas. H&E and immunostaining for RKIP, PIRIN, BCL2, BCL3, MITF, and ANXA5 in representative primary melanoma biopsies.

**Figure 2 cancers-15-02174-f002:**
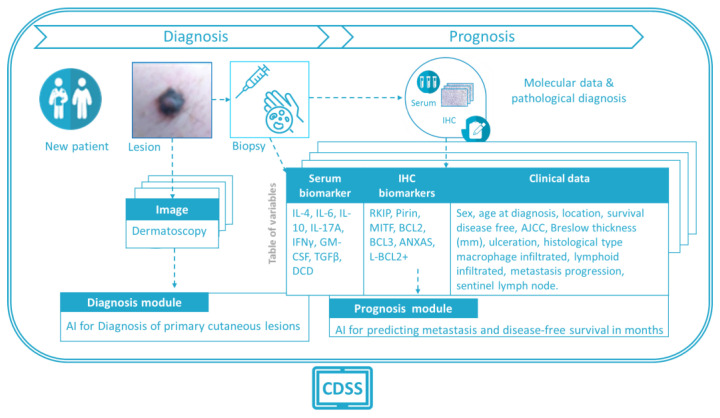
Scheme of the proposed workflow used to incorporate the demographic and clinical data, the lesion images, and new molecular biomarkers into the AI models for melanoma diagnosis and prognosis.

**Figure 3 cancers-15-02174-f003:**
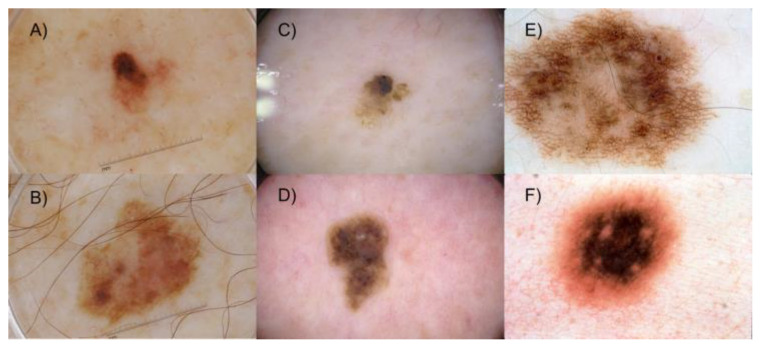
Representative dermatoscopy images of melanoma (**A**,**B**), seborrheic keratosis (**C**,**D**), and other lesions (**E**,**F**) from the ISIC 2017 challenge dataset.

**Figure 4 cancers-15-02174-f004:**
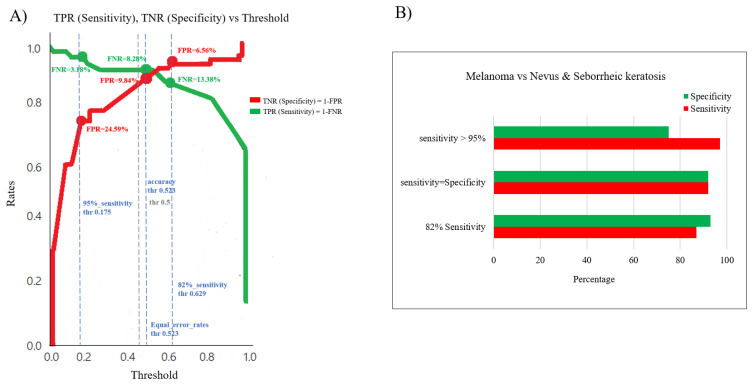
Performance of the CDSS’s diagnostic module for the primary diagnosis of melanoma based on clinical images. (**A**) Sensitivity (TPR) and specificity (TNR) were compared for the different sensitivity thresholds. (**B**) Corresponding ROC curves of the classification models for the sensitivity and specificity values reported in A.

**Figure 5 cancers-15-02174-f005:**
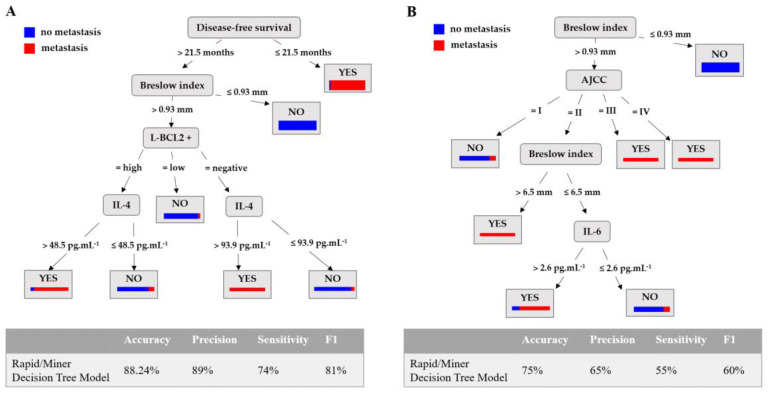
The RapidMiner decision tree model used to predict the development of metastasis. (**A**) The performance of the models obtained to predict the development of metastasis and the set of rules are shown in Appendix A: disease-free survival is expressed in months; the Breslow Index in mm; L-BCL2^+^ represents the quantification of the intratumoral BCL2-positive lymphocytes expressed as No (absence of BCL2^+^ intratumoral lymphocytes), Low (presence of a small number of BCL2^+^ intratumoral lymphocytes), and High (presence of a large number of BCL2^+^ positive intratumoral lymphocytes); serum IL-4 is expressed in pg·mL^−1^. (**B**) The RapidMiner decision tree model trained with the original 196 patients but without the attribute disease-free survival (see rules in Appendix A). The attributes used to predict metastasis are Breslow index, stage according AJCC, and IL-6-serum levels in pg·mL^−1^.

**Figure 6 cancers-15-02174-f006:**
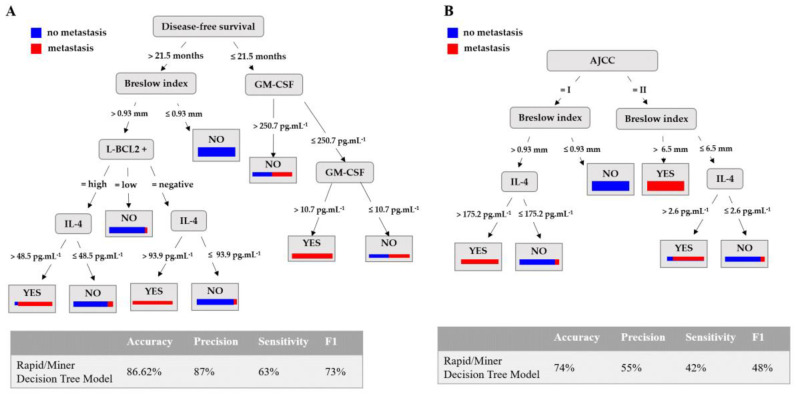
RapidMiner decision tree model to predict metastasis in early-stage melanoma patients (stages I and II). (**A**) The performance of the models obtained to predict the development of metastasis and the set of rules are shown in Appendix A: disease-free survival is expressed in months; the Breslow Index in mm; L-BCL2^+^ represents the quantification of the intratumoral BCL2^+^ lymphocytes expressed as No (absence of BCL2^+^ intratumoral lymphocytes), Low (presence of small numbers of BCL2^+^ intratumoral lymphocytes), and High (presence of large numbers of BCL2^+^ intratumoral lymphocytes); and serum GM-CSF and IL-4 expressed in pg·mL^−1^. (**B**) The RapidMiner decision tree model trained with the original 149 patients but without the attribute disease-free survival (see rules in Appendix A). The attributes used to predict metastasis are stage according AJCC Breslow index, and IL-4 and IL-6 serum levels.

**Table 1 cancers-15-02174-t001:** Clinical data of melanoma patients.

	Nº	(%)
**Melanomas**	196	
Age at diagnosis	58 (range, 23–88)	
**Sex**		
Male	88	(45)
Female	108	(55)
**Disease Evolution**		
Disease-free	131	(67)
Metastasis	65	(33)
**Localization**		
Head and Neck	34	(17)
Trunk	63	(32)
Upper limb	19	(10)
Lower limb	58	(30)
Acral	14	(7)
Others	5	(2.5)
Unkown	3	(1.5)
**Histological subtype**		
SSM	119	(61)
NM	46	(23.5)
ALM	14	(7)
LMM	9	(4.5)
LM	2	(1)
Others	6	(3)
**AJCC stage at diagnosis**		
In situ	26	(13)
I	92	(47)
II	57	(29)
III	17	(9)
IV	4	(2)

SSM (Superficial Spread Melanoma), NM (Nodular Melanoma), ALM (Acral Lentigo Melanoma), LMM (Lentigo Malignant Melanoma), LM (Lentigo Melanoma).

## Data Availability

The data can be shared up on request.

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
