# Peer review of "Melanoma Clinical Decision Support System: An Artificial Intelligence-Based Tool to Diagnose and Predict Disease Outcome in Early-Stage Melanoma Patients"

_cancers, 2023, doi:10.3390/cancers15072174_

Round 1

Reviewer 1 Report

ln 210. There is little information about how this model was trained, and its design, e.g. layers, etc. either in the text or in the reference.

ln 214. again, little detail was revealed about how model was refined with 196 collected cases. I assumed that the 196 cases were separate from the 1600 images used to train the model. There is a problem as pointed out by the author in line218-219. e.g., illumination and color, etc. ,which could severely impact model.

ln 231. the prognostic module requires 28 attributes to work. How to deal with situations where less attributes were collected? Would the model still hold?

line 233. different configurations and parameters...." needs further clarification. This is too generic for scientific publication. 

line 237. 10-fold cv was mostly used for validation during training of models, it does not test the ability of model to predict new data. Only an independent test set could do that.

line 243-248. why there is a need to define sensitivity and specificity here?

line 252. how the best thresholds were determined?

line 259-269. the paragraph is quite confusing, please consider revise it.

line 280. it is more likely that the authors have not optimize the training of such classifiers, which resulted in close to 0% sensitivity. Usually random forest works pretty well with default parameters. Again, missing information here. the training set is not poorly balanced either. so something not right with the training of classifieres.

line 286. Again, why all the definitions of evaluation terms?

line 300. I have concerns over using disease-free survival as a predictor. It is an outcome that can't be directly used to assess patients when they first presented for diagnosis.

Reviewer 2 Report

Diaz-Ramon and colleagues show a decision support system (DSS) for the diagnosis and prognostic evaluation of primary melanoma. This DSS is based on the analysis of serological and histopathological biomarkers with artificial intelligence algorithms and can be used for the prediction of the risk of metastasis and estimation of the disease-free interval of melanoma patients.

This is a very timely study as the integration of these systems in clinical practice can help alleviate the heavy workload of pathologists while assisting inexperienced pathologists in making informed decisions.

The work presented here has significant strengths but also some weaknesses. The main concern of the reviewer is whether a more in-depth analysis of the obtained results will reveal weaknesses in the presented study, therefore, affecting the claims performed by the authors. Specifically:

    • Introduction. There is no motivation behind the selection of serological or histopathological biomarkers. Why these markers and no others? More background would help the general reader to understand the choice of biomarkers.

    • In section 2.3.1, diagnostic module, the authors train an initial model using 1600 clinical images of nevi, melanoma, and seborrheic keratosis. This model is later fine-tuned with 196 cases of melanoma splitting samples between training, validation, and testing to evaluate the model. Does this splitting of samples refer to the 196 melanoma cases only? If so, what is the performance of the model in non-melanoma cases? Perhaps the authors can leave a fraction of the initial 1600 images out, and evaluate the performance of the model after fine-tuning with melanoma images. Otherwise, the model will be biased toward melanoma predictions.

    • For patients developing metastasis and undergoing treatment, did the authors consider treatment differences? Are all the patients treated the same way? Explain.

    • In Table 1 it is clear that some histological subtypes of melanoma (LM, LMM, ALM) are under-represented. How do the different modules perform when segmenting for the clinical variables described in table 1? Can the authors generalize their tool for any type of melanoma? Or, is there a performance difference based on clinical factors?

    • It is unclear how the diagnostic model performs for people with different skin colors.

    • There is no mention of which architecture was used for the CNN model(s).

    • Lines 266-269 → This fits better in the discussion section.

    • Lines 289-291 → Perhaps the authors want to use Matthew’s correlation coefficient if the goal is to consider the issue of unbalanced datasets.

    • Figure 5 & Figure 6. The parent node of the decision tree is disease-free survival. However, does it make sense to include DFS in the model? How does this extrapolate for the application of the model on newly diagnosed samples? 

Author Response

Olease see the attachment

Reviewer 3 Report

Cancers - 2234461

Melanoma Clinical Decision Support System: an artificial intelligence-based tool to diagnose and predict disease outcome in early-stage melanoma patients

Díaz-Ramón JL1,2,#, Gardeazabal J1,2,#, Izu R2,3,#, Garrote E4,5, Rasero J6, Apraiz A2,5, Penas C2,5, Seijo S7, Saratxaga CL4, de la Peña PM7, Sánchez-Diez A2,3,, Cancho-Galán G2,3, Velasco V1,2, Sevilla A2,10, Fernández D11, Cuenca I7, Cortés J2,5,12, Alonso S10, Asumendi A2,5,* and Boyano MD2,5,*

Summary

In this manuscript, Diaz-Ramon et al., introduce a machine learning approach to predict disease outcome in early-stage melanoma patients. In general, the manuscript is written straightforward and the results are presented in a very clear style.

Hence, I have some concerns:

Major 

The AI is trained with two datasets (lines 210 and 215), nevertheless the Data Availibility Statement (l. 512) is declared as N/A. The authors should complete this statement according to the information provided anyway in their manuscript and declare, how both data sets can be retrieved. Is it possible, for instance, to provide the final dataset of 196 patients x 28 attributes for other researchers? Furthermore, I miss a flow chart supporting the explanations in line 318ff on the selection and removal of the introduced 196 patients.

I missed a discussion about the state-of-the-art on machine learning and melanoma diagnosis/prognosis and a comparison to competitive work in this field. References #9-11 refer and discussed their previous work, while I was not able to detect reference #12 in the manuscript. Considering the title of #12 it may be relevant to address my issue. Further, a quick Pubmed search finished with approx. 50 publications since 2015 related to the introduced research topic. Three examples can be found above, therewith I would suggest to extend the introduction. Of course, the authors are free to discuss other publications than the three examples.

Finally, a key element of the introduced model is a correction tool for illumination and lesion segmentation that overcomes several known limitations in image analysis. While I appreciate the authors efforts to eliminate such sources of errors, I am amazed about the fact that this tool is available since 2017 in Arxiv only. The authors describe this tool in short as “proved to be very useful”. This wording in this context is a suggestive of a peer-reviewed process, whom the referenced manuscript #16 never passed. I recommend to extend this statement with a short introduction to this tool, an explanation about the mentioned approval and a discussion of pros and cons of the applied illumination correction. Are similar correction tools available? What is the advantage of the applied tool against the others.

The latter criticism follows a clear policy, that the use of not peer-reviewed data and manuscripts in a scientific presentation requires a strict explanation and a noticeable differentiation to peer-reviewed content.

Pubmed results

Pham, TC., Luong, CM., Hoang, VD. et al. AI outperformed every dermatologist in dermoscopic melanoma diagnosis, using an optimized deep-CNN architecture with custom mini-batch logic and loss function. Sci Rep 11, 17485 (2021). https://doi.org/10.1038/s41598-021-96707-8

Jojoa Acosta, M.F., Caballero Tovar, L.Y., Garcia-Zapirain, M.B. et al. Melanoma diagnosis using deep learning techniques on dermatoscopic images. BMC Med Imaging 21, 6 (2021). https://doi.org/10.1186/s12880-020-00534-8

Tyler Safran, Alex Viezel-Mathieu, Jason Corban, Ari Kanevsky, Stephanie Thibaudeau, Jonathan Kanevsky, Machine learning and melanoma: The future of screening, Journal of the American Academy of Dermatology, Volume 78, Issue 3, 2018, Pages 620-621, ISSN 0190-9622, https://doi.org/10.1016/j.jaad.2017.09.055.

Minor

1.     Manuscript title: The title announces an AI tool on diagnosis, which is presented in section 3.1. However, it is not clear to me, how the diagnosis AI can be applied and which benefits can be achieved with this tool considering the main object of disease outcome prediction.

2.     The authors defined a target problem during classification (line 213ff) and explain later in the results, that the unbalanced data may bias the success of other models (lines 280ff). As CNN’s are known to be vulnerable for bias from unbalanced sets too it would be helpful to know how the authors avoid this issue for the introduced CNN.

3.     The authors introduce a dataset covering 28 attributes. Do I understand this correctly that the final RapidMiner Model took advantage of an attribute subset, mentioned in figures 5 and 6? If yes, I would suggest to emphasize this reduction, if no, please clarify how these 28 attributes are applied in RapidMiner.

4.     RapidMiner was determined to be the best model (lines 239 and 343). While the F1-score was selected for the prediction model, which score is applied for the prognostic module? In addition, I would suggest to present a short overview on the stats of the best 3 or likewise models analogous to the tables in figure 5 and 6. It is unclear, whether RapidMiner won a tough competition or outperforms any other model.

5.     The DT’s of RapidMiner are presented in Figure 5 and Figure 6. Do the other models use the same DT’s or did they generate individual trees during training?

6.     Is the stratification to subsets during the 10-fold cross-validation (line 239) identical for each model or was the stratification applied individually to each model?

7.     Can the authors quantify the impact of the unbalanced dataset (line 283)? In other words: What is the model outcome, if you reduce the larger set to balance both datasets?

8.     Does the model require dermatoscopy images for prediction or can it predict the disease outcome based on the 28 attributes only? In this case, how are specificity and sensitivity affected?

9.     Was the model applied in daily clinical routine? Which differences in treatment result from the model prediction? What is the take-home message: Can clinicians apply this tool to support their own work?

10. Figure 4A: Please improve the image quality focussing on readability.

11. Figure S1A: Visualization can be improved, maybe through larger gaps between the model bars. It is hard to distinguish which bar correspond to which model.

  1. Figure S1B and S1D: It seems that both figures contain the DT’s presented in Figure 5 and Figure 6. If this is your intention, please add a caption note pointing to Fig 5 and Fig 6.

Round 2

Reviewer 1 Report

Thank you for making necessary changes, I have no further comments

Reviewer 2 Report

The authors have addressed all the comments suggested by the reviewer

Reviewer 3 Report

The authors addressed most of my concerns sufficiently.

I am wondered, why the Data Availibity Statement is still N/A.

And I remark, that the relation between ref #20 and #21 does not appear during reading. For readers from other communities, it would be helpful to point on this connection in the main text, resolving my criticism on the Arxiv manuscript.